# Mass Cytometric Analysis of Early-Stage Mycosis Fungoides

**DOI:** 10.3390/cells11071062

**Published:** 2022-03-22

**Authors:** Nannan Guo, Li Jia, Coby Out-Luiting, Noel F. C. C. de Miranda, Rein Willemze, Frits Koning, Maarten Vermeer, Koen Quint

**Affiliations:** 1Department of Immunology, Leiden University Medical Center, 2300 RC Leiden, The Netherlands; n.guo@lumc.nl (N.G.); l.jia@lumc.nl (L.J.); f.koning@lumc.nl (F.K.); 2Department of Dermatology, Leiden University Medical Center, 2300 RC Leiden, The Netherlands; j.j.out@lumc.nl (C.O.-L.); r.willemze.huid@lumc.nl (R.W.); m.h.vermeer@lumc.nl (M.V.); 3Department of Pathology, Leiden University Medical Center, 2300 RC Leiden, The Netherlands; n.f.de_miranda@lumc.nl

**Keywords:** mass cytometry, CyTOF, imaging mass cytometry, IMC, mycosis fungoides, cutaneous T-cell lymphoma, CTCL

## Abstract

Mycosis fungoides (MF) is the most common subtype of cutaneous T-cell lymphoma. Early-stage disease is characterized by superficial infiltrates of small- to medium-sized atypical epidermotropic T lymphocytes that are clonal related. Nevertheless, the percentage of atypical T cells is low with many admixed reactive immune cells. Despite earlier studies, the composition and spatial characteristics of the cutaneous lymphocytic infiltrate has been incompletely characterized. Here, we applied mass cytometry to profile the immune system in skin biopsies of patients with early-stage MF and in normal skin from healthy individuals. Single-cell suspensions were prepared and labeled with a 43-antibody panel, and data were acquired on a Helios mass cytometer. Unbiased hierarchical clustering of the data identified the major immune lineages and heterogeneity therein. This revealed patient-unique cell clusters in both the CD4^+^ and myeloid cell compartments but also phenotypically distinct cell clusters that were shared by most patients. To characterize the immune compartment in the tissue context, we developed a 36-antibody panel and performed imaging mass cytometry on MF skin tissue. This visualized the structure of MF skin and the distribution of CD4^+^ T cells, regulatory T cells, CD8^+^ T cells, malignant T cells, and various myeloid cell subsets. We observed clusters of CD4^+^ T cells and multiple types of dendritic cells (DCs) identified through differential expression of CD11c, CD1a, and CD1c in the dermis. These results indicated substantial heterogeneity in the composition of the local immune infiltrate but suggest a prominent role for clustered CD4–DC interactions in disease pathogenesis. Probably, the local inhibition of such interactions may constitute an efficient treatment modality.

## 1. Introduction

Mycosis fungoides (MF) is the most common variant of cutaneous T-cell lymphoma (CTCL), with a 5 year disease-specific survival of approximately 80%, depending on the stage of the disease [1,2]. The majority of MF patients have early-stage disease (Stage IA/IB), with red scaly (sometimes itchy) patches and plaques on the skin with no extracutaneous involvement. Early stage MF can be extremely subtle, with skin lesions only present on a small portion of the body’s area. Those lesions are hard to recognize by untrained physicians and can be easily confused clinically as well as histologically with reactive dermatitis, leading to a diagnostic delay and mistreatment [3,4]. Definite diagnosis is made by clinicopathological correlation and is sensitive to error. Therefore, the need for adjuvant diagnostic tools is desirable.

Histologically, early-stage MF is characterized by small- to medium-sized pleomorphic T-cell lymphocytes with cerebriform nuclei infiltrating the basal layer of the epidermis (epidermotropism) [5]. However, the characteristic tumor cells in early-stage MF can be scarce and hard to recognize. Immunophenotyping, to document the loss of T-cell antigens suggestive of a malignancy (CD3, CD5, and CD7), and to determine the phenotype of the malignant cells (CD4, CD8, and cytotoxic markers), can be of help. Still the diagnosis of early-stage MF remains challenging, since early-stage MF lesions are characterized by a low number of tumor cells and an extensive inflammatory infiltrate with a relative abundance of CD8^+^ T cells and dendritic cells (DCs). Multiple studies suggest that complex loops of cytokine and chemokine signaling between fibroblasts, keratinocytes, and malignant T cells contribute to an inflammatory microenvironment that contributes to the progression of disease. These studies suggested that a decrease in CD8^+^ T cells and an increase in inhibitory M2 macrophages, combined with a shift from a Th1- to a Th2-dominated microenvironment and alteration of the regulatory T-cell (Treg)/Th17 balance leads to progression of MF. Furthermore, the viability and growth of the malignant T cells seemed partly dependent on interaction with immature dendritic cells, through contact between CD40 located on dendritic cells and the CD40 ligand located on malignant T cells [6]. Another in vitro study further supported previous observation in which dendritic cells stimulates cultures of malignant T cells and induces Treg cytokine production [7]. However, comprehensive studies investigating the different subsets of immune cells in tissue context are largely lacking, and little is known about cell–cell interactions between tumor cells and the reactive immune infiltrate.

Recently, single-cell phenotyping platforms, such as single-cell RNA sequencing and mass cytometry (cytometry by time-of-flight; CyTOF), have been utilized to investigate cellular heterogeneity [8,9], to identify novel cellular subsets [10,11], and to discover clinically relevant biomarkers with clinical value in humans [12]. Mass cytometric techniques can characterize different immune cell populations in dissected skin cells and in the tissue contexture [13,14]. Single-cell mass cytometry allows for an unbiased analysis of the complexity and heterogeneity of the human immune system, as over 40 unique cellular markers can be measured simultaneously [15,16]. In addition, imaging mass cytometry, which couples a laser ablation system with a mass cytometer, allows for the analysis of up to 40 markers in a single tissue section. With imaging mass cytometry, tissue structures can be visualized as well as the composition, distribution, and spatial interactions of the stromal and immune cell subsets in the tissue context [17,18].

In the current study, we applied single-cell suspensions and imaging mass cytometry to skin biopsies of early-stage MF patients to gain deeper insight into the composition and organization of the immune compartment in the tissue context and the relation/interaction with malignant T cells. We observed pronounced patient-specific characteristic features in the skin resident immune populations, and distinct clusters of lymphoid and myeloid cell populations in lesioned skin that may drive the transformation and subsequent sustenance of the tumor cells in situ.

## 2. Material and Methods

### 2.1. Patient Selection

A total of 16 skin biopsies (4 mm) with confirmed diagnoses of early-stage MF (stage IA/IB; patches/plaques, no tumors) and 21 biopsies of normal skin (NS) were selected from the “Biobank Dermatology” of the Leiden University Medical Center. All biopsies were taken from the lesions of MF patients that were not previously treated with topical steroids, and the MF patients had not yet received any systemic medication at the time of biopsy. All diagnoses were confirmed by an expert panel of dermatologists and pathologists of the Dutch Cutaneous Lymphoma Group. An additional biopsy from the same lesion and photographic documentation were available for routine diagnostics and determination of the subtype of MF (i.e., folliculotropic MF, CD8^+^ MF). Approval by the medical ethical commission of the Leiden University Medical Center (protocol: B19.005) was obtained in accordance with the local ethical guidelines and the Declaration of Helsinki. All patients provided written informed consent for biobanking. Baseline characteristics and disease course during follow up are given in Appendix A, and the sex and age characteristics of the NS donors are shown in Appendix A.

### 2.2. Skin Biopsy Processing

The fresh skin biopsies (MF, *n* = 10; NS, *n* = 17, for suspension mass cytometry) and the snap-frozen biopsies (MF, *n* = 6; NS, *n* = 4, for imaging mass cytometry) were retrieved from the biobank. Fresh skin punch biopsies were maintained in cold HBSS solution and brought to the laboratory within 10–30 min. To obtain single-cell suspensions, the skin biopsies were cut into small pieces and transferred to a gentleMACS C tube to incubate in 500 μL IMDM (Lonza, Basel, Switzerland) supplemented with 10% FCS, 1 mg/mL collagenase D (Roche Diagnostics), and 50 μg/mL DNase I (Roche Diagnostics, Basel, Switzerland) at 37 °C for 2 h, after which 500 μL of IMDM with 10% FCS was added to terminate digestion. Subsequently, the gentleMACS program h_skin in gentleMACS™ Dissociator (Miltenyi Biotec, Bergisch Gladbach, Germany) was run, after which the cells were spun down. Finally, cell suspensions were filtered through a 70 μm nylon cell strainer and immediately stained with the single-cell mass cytometry antibody panel. Snap-frozen skin tissues were obtained by embedding in optimal cutting temperature compound (O.C.T., VWR), frozen in cold isopentane (VWR), and stored at −80 °C for immunodetection staining by IMC.

### 2.3. Suspension Mass Cytometry Antibody Staining and Data Acquisition

Metal-conjugated antibodies used for single-cell mass cytometry are listed in Appendix A. For self-conjugation of antibodies, BSA-free and carrier-free formulations of antibodies were purchased from different suppliers. Subsequently, conjugation of antibodies with lanthanide metals was performed using the Maxpar Antibody Labeling Kit (Fluidigm, San Francisco, CA, USA) following the manufacturer’s instructions. Post-conjugation, all antibodies were eluted in 100 μL W-buffer (Fluidigm) and 100 μL antibody stabilizer buffer (Candor Bioscience) supplemented with 0.05% sodium azide. Procedures for mass cytometry antibody staining and data acquisition were performed as previously described [19]. In short, skin cells were incubated with 1 mL 250 nM Cell-ID™ intercalator-103Rh (Fluidigm) for 15 min at room temperature (RT) to distinguish live cells from dead cells in 5 mL microcentrifuge tubes. After washing once by Maxpar^®^ Cell Staining Buffer (Fluidigm), the skin cells were stained with metal-conjugated antibodies for 45 min at RT (Appendix A). After staining and washing by Maxpar^®^ Cell Staining Buffer (Fluidigm) three times, cells were incubated with 1 mL 2000× diluted 250 μM Cell-ID™ intercalator-Ir (Fluidigm) in Maxpar Fix and Perm Buffer (Fluidigm) overnight at 4 °C. The next day, skin cells were washed by Maxpar^®^ Cell Staining Buffer (Fluidigm) 3 times and spun down. Finally, cells were acquired on a Helios time-of-flight mass cytometer (Fluidigm). Data were normalized using EQ Four Element Calibration Beads (Fluidigm) with the reference EQ passport P13H2302 in each experiment.

### 2.4. Imaging Mass Cytometry Staining and Data Acquisition

Procedures for IMC antibody staining and data acquisition for snap-frozen tissue were carried out as previously described [20]. In short, snap-frozen human skin biopsies were sectioned at a thickness of 5 μm. All tissue sections were dried for 1 h at RT after cutting; then, they were fixed by incubating with 1% paraformaldehyde (PFA) for 5 min at RT followed by 100% methanol for 5 min at −20 °C. After fixation procedures, tissue sections were washed in Dulbecco’s phosphate-buffered saline (DPBS, ThermoFisher Scientific, Waltham, MA, USA) with 1% bovine serum albumin (BSA, Sigma) and 0.05% Tween, and rehydrated in additive-free DPBS. After washing, the tissue sections were blocked by Superblock Solution (ThermoFisher Scientific) for 30 min in a humid chamber at RT. Tissue sections were then stained with a 36 antibody mixture overnight at 4 °C; all antibodies in the IMC panel are listed in Appendix A. After the antibody mixture incubation, the tissue sections were washed and incubated with 125 nM Cell-ID™ Intercalator-Ir for 30 min at RT. After an additional wash, tissue sections were washed with Milli-Q water (Merck Millipore, Burlington, MA, USA) for 1 min to remove additives and dried for 20 min at RT. The acquisition was performed by UV-laser, spot by-spot, using a Hyperion Imaging System at a resolution of 1 μm and a frequency of 200 Hz. Regions of interest (ROIs) with 1000 × 1000 μm or 1200 × 1000 μm were selected in skin tissue sections. We ablated the whole skin tissue section by 1–2 ROIs to cover the whole skin biopsy sections. After ablation by Hyperion (Fluidigm), as described in the Hyperion Imaging System’s user guide, MCD files and .txt files were generated for each sample for further analysis.

### 2.5. Data Analysis

Data for single, live CD45^+^ cells were gated from each MF sample individually using FlowJo software as shown in Appendix A. Subsequently, the data were sample tagged and hyperbolic arcsinh transformed with a cofactor of 5 and subjected to dimensionality reduction analysis in Cytosplore^+HSNE^ [15,16]. We employed a 43-antibody panel for staining of single-cell mass cytometry (Appendix A). Major immune lineages were identified at the overview level of a hierarchical stochastic neighbor embedding (HSNE) analysis on CD45^+^ cells’ data from all samples with default perplexity (30) and iterations (1000). All HSNE and t-SNE plots were generated in Cytosplore [15,16]. The immune lineage population frequencies of CD45^+^ cells were computed within the individual samples using the “prcomp” function, and the result was visualized using the “ggbiplot” function in R software. Hierarchical clustering of the phenotype heatmap was created with Euclidean correction and average linkage clustering, while the cell frequency heatmap with Spearman correction and average linkage clustering was generated in MATLAB R2016b.

We utilized a 36-antibody panel for staining of IMC on snap-frozen skin tissue to visualize spatial data (Appendix A). All IMC raw data were from three independent experiments and analyzed using the Fluidigm MCD™ viewer (v1.0.560.2). Images of single markers within the IMC panel are shown in Appendix A to show the individual stains in each NS and MF sample, and the minimum and maximum threshold of each marker for all samples are provided in Appendix A in which the maximum threshold value reflects the staining density of the antibodies, while the minimum threshold value was used to reduce the background signal for each marker channel. To combine related markers to visualize the structure of skin tissue and distinct immune subsets, we utilized the Fluidigm MCD™ viewer to generate the images for a single ROI of skin tissue.

## 3. Results

### 3.1. Identification of Major Immune Lineages

We applied a 43-antibody panel to identify the major immune lineages (CD4^+^ T cells, CD8^+^ T cells, myeloid cells, B cells, and innate lymphoid cells (ILCs)) in 10 early-stage MF patients. This antibody panel contained markers for the identification of the immune lineages, cellular differentiation, activation, trafficking, tissue residency, and function (Appendix A). Single, live CD45^+^ cells were selected by means of DNA and CD45 staining, and commonly used mass cytometry parameters (Appendix A). From the skin biopsies, we acquired, on average, 1737 CD45^+^ cells from normal skin (NS) and 16,396 CD45^+^ cells from MF skin by single-cell mass cytometry (Figure 1A). In agreement, IMC analysis revealed substantial immune cell infiltration in situ in MF patients in comparison with NS samples (Figure 1B). Due to the much fewer number of immune cells detected in the NS samples by both techniques, we next focused on MF patients for further analysis.

To analyze the composition of the immune cell infiltrate of the MF patients, we integrated the data derived from 10 MF samples (3.6 × 10^5^ CD45^+^ cells) and performed an HSNE analysis in Cytosplore^+HSNE^ at the global level to identify the major immune lineages (Figure 2A). Based on the marker expression profiles (Figure 2B) and density features of the embedded cells (Appendix A), we identified clusters of CD3^+^CD4^+^ T cells, CD3^+^CD8^+^ T cells, CD11c^+^/CD11b^+^ myeloid cells, CD3^−^CD20^–^CD11c^−^CD11b^−^CD7^+^ innate lymphoid cells (ILCs), CD20^+^ B cells, and abnormal CD3^−^CD7^−^CD8^−^CD4^+^ T cells (Figure 2A and Appendix A). Moreover, we quantified the relative frequencies of these major immune lineages within the CD45^+^ cells of each MF patient (Figure 2C). We observed that CD3^+^CD4^+^ T cells were the dominant cell population in MF patients, as they represented more than 50% of CD45^+^ cells in eight out of ten MF patients. Both CD3^+^CD8^+^ T cells and myeloid cells were present in all patients, but their percentage varied significantly among patients. In five out of 10 patients, aberrant T cells with loss of CD3 and CD7 expression (CD3^−^CD7^−^CD8^−^CD4^+^ T cells) were observed. This aberrant phenotype was most prevalent in patient 79MF but lower in the other four patients (i.e., 84MF, 82MF, 81MF, and 62MF). Significant numbers of CD3^−^CD20^−^CD11c^−^CD11b^−^CD7^+^ ILCs were only detected in three patients, while only few B cells were present in all patient samples. Together, these global analyses revealed that all major immune lineages could be readily identified in MF patients, and the composition of these immune cells differed among the evaluated patients.

### 3.2. Analysis of the CD4 T-Cell Compartment Revealed Shared and Patient-Unique Features

To define and compare the composition of the T-cell immune compartments among the MF samples, we next selected the T-cell clusters individually and performed a tSNE analysis at the single-cell level. The embedding of the CD3^+/−^CD4^+^ T cells indicated that next to shared features (encircled in gray, Figure 3A), distinct clusters of cells that were highly enriched in individual patients were also readily identified (encircled in black, Figure 3A). Based on the density features and related marker expression profiles of the t-SNE-embedded cells (Figure 3B and Appendix A), we identified 27 distinct CD4^+^ T-cell clusters, each defined by a unique marker expression profile (Figure 3C). The associated cell frequency heatmap gives an overview of the relative abundance of those subsets in the patient samples (Figure 3D).

Unsupervised hierarchical clustering of the patient samples based on the cell frequency heatmap grouped six samples together, while the other four samples were distinct (Figure 3D, top). Visual inspection of the heatmap indicated that the clustering of the six samples was, to a large extent, due to the sharing of clusters CD4 T-2, CD4 T-6, and CD4 T-7 and, to a lesser extent, by CD4 T-1 and CD4 T-4 (Figure 3D, pink boxes), which was confirmed by the actual percentage of these subsets in the individual patient samples (Figure 3E). The actual percentage of these subsets in the individual patient samples support the notion that cluster CD4 T-1 and CD4 T-4 (CD3^+^CD5^+^CD127^+^CD45RO^+^CCR7^−^ Effector memory) and CD4 T-2 (CD3^+^CD5^+^CD127^−^CD45RO^+^CD25^+^ Treg-like) were the most important in this respect (Figure 3E). In contrast, the remaining four patient samples were distinguished by the presence of distinct clusters of CD4 T cells (Figure 3D, black boxes) that were either CD3^+^ or CD3^–^, many of which displayed variable expression levels of CD30 and CLA (Figure 3C) and, thus, likely contained the abnormal cells in these patients. Here, patient 62MF was typified by the presence of CD4 T-26 and T-27 (CD3^−^CD103^+/−^CD26^+^CD30^low^CD5^+^CD127^+^PD1^+^CD69^+^CD45RO^+^); patient 71MF by cluster CD4 T-15, T-18, and T-28 (CD3^+^CD103^+/−^CD30^low^CLA^low^CD5^+^PD1^+^CD69^+^CCR4^+^CCR7^+^CD28^+^CD45RO^+^); patient 79MF by cluster CD4 T-11 and T-14 (CD3^+/−^CD26^+^CD30^+^CLA^+^CD25^low^CD5^+^CD127^+/−^CD69^+^CD28^+^CD45RO^+^) and CD4 T-16 (CD3^−^CD26^+^CD5^−^CD127^−^CD69^+^CD28^−^CD45RO^+^); patient 59MF by cluster CD4 T-23 and T-24 (CD3^−^CD161^+^CD26^+^CD30^+/−^CLA^+/−^CD127^+^CD103^+/−^PD1^+^CD69^+^CCR6^+^CD45RO^+/−^) and cluster CD4 T-21 and CD4 T-22 (CD3^+^CD161^+^CD103^low/−^CD30^−^CD127^low^CLA^+/−^CCR4^+^CD28^−^CCR7^+^) (Figure 3C). The actual percentages of these cell clusters in the individual patients underscores that these were the dominant cell types in the investigated biopsies (Figure 3F). In particular, the dominance of the clusters in patients 71MF, 79MF, and 59MF is striking, where patient 79MF is distinct, as they were the only one harboring a large population of the CD3^−^CD30^+^CD5^+^CD4^+^ T cell (cluster CD4 T-11) (Figure 3D,F).

A similar analysis of the CD8^+^ T cells showed that the distribution CD8^+^ T cells was highly similar in nine out of ten patients (Appendix A). Based on marker expression profiles and density feature, 25 CD8^+^ T cell clusters were identified (Appendix A). We observed that hierarchical clustering resulted in three patient groups (Appendix A), while 71MF patient was separate due to the high proportion of cluster CD8 T-26 and T-27 (CD103^+^CLA^+^PD-1^+^) (Appendix A).

In conclusion, hierarchical clustering of the T-cell compartments revealed both shared and unique features of the MF patients, where the shared features may relate to an earlier disease state, while the appearance of highly patient-specific cell clusters is probably related to evolution and the response of the tumor microenvironment.

### 3.3. Analysis of the Myeloid Compartment Revealed Shared and Patient-Unique Features

We next analyzed the myeloid compartment. Similar to the CD4^+^ T cells, the tSNE analysis revealed the presence of subsets that were overrepresented in particular MF patients (Figure 4A; patients 26MF, 62MF, 71MF, and 84MF, encircled). Based on the distribution of the marker expression profiles (Figure 4B) and the density features of the t-SNE-embedded cells (Appendix A), 28 distinct myeloid cell clusters were identified that fell within six major clusters (CD11c^dim^HLA-DR^+^ myeloid cells, CD11c^−^HLA-DR^+^ cells, CD14^+^ monocytes, CD16^+^CD15^dim^ monocytes, HLA-DR^+^CD1a^+^dendritic cells, and CD163^+^ macrophages) (Figure 4C).

Compared to the CD4^+^ T cells, the myeloid cell-associated cell frequency heatmap was much more complex and indicative of much heterogeneity (Figure 4D). However, the subset Mye-3 (CD11b^low^CD11c^+^CD14^dim^HLA-DR^+^CD4^low^) was present in all MF patients (Figure 4D, pink box), and cluster Mye-11 (CD11b^−^CD11c^+^HLA-DR^+^CD1a^dim^CD4^low^) was also observed in the majority of MF patients (Figure 4D, pink box, and Figure 4E), while cluster Mye-19 (CD1a^+^CD11b^+^CD11c^+^HLA-DR^+^) marked patients 79F, 59MF, and 66MF (Figure 4D, black box, and Figure 4F). In contrast, cluster Mye-28 (CD11b^+^CD11c^−^CD14^+^CD45RA^+^CD163^−^HLA-DR^−^) was essentially only present in patient 26MF and cluster Mye-16, 18, and 23 (CD11b^low/−^CD11c^+^ HLA-DR^+^CD163^+/−^) only in 71MF (Figure 4D, black boxes and Figure 4F). Finally, patient 82MF and 84MF shared cluster Mye-22 (CD11b^−^CD11c^−^CD14^+^HLA-DR^+^CD163^+^CD4^low^), and patient 69MF and 81MF shared cluster Mye-6 (CD11b^−^CD11c^−^HLA-DR^+^CD1a^+^CD4^low^) (Figure 4D, black boxes, and Figure 4F). Thus, similar to the CD4^+^ T-cell compartment, the analysis of the myeloid compartment also revealed both shared and unique features of the MF patients that may relate to different stages of disease progression. Here, it is important to note that, in some patients. (71MF, 79MF, and, to a lesser extent, 69MF and 59MF) both patient-unique CD4^+^ T and myeloid cell populations were observed.

### 3.4. Imaging Mass Cytometry Revealed Spatial Signatures In Situ

To determine the spatial distribution of the immune and stromal cells in situ, we applied an IMC panel comprising 36 antibodies to tissue sections of NS and MF skin biopsies. The antibody panel contained markers to visualize the overall tissue architecture such as E-cadherin (epithelium), vimentin (intermediate filament), α-smooth muscle actin (αSMA) and collagen I (extracellular matrix), and markers to identify T cells (i.e., CD3, CD8, CD4, CD25, FOXP3, CD45RA, and CD45RO), B cells (CD20), NK cells (i.e., CD7 and CD56), myeloid cells (CD11c), dendritic cells (i.e., CD1a, CD1c, CD123, and CD141), mast cells (i.e., CD117 and FcεRIα), monocytes and macrophages (i.e., CD14, CD68, CD163, and HLA-DR). In addition, CD31 was included to identify endothelial cells and Ki-67 to identify cell proliferation (Appendix A). With this panel, snap-frozen tissue sections derived from four NS controls and six early-stage MF patients were stained after which data were acquired using Hyperion Imaging Mass Cytometry.

To obtain further information on the phenotype and localization of the immune cell subsets within the tissue context, we visualized the data by combinations of specific markers (Figure 5 and Figure 6). Here, the combination of E-cadherin and DNA as nuclear counterstain were used to distinguish the epidermis from the dermis on skin tissues from the NS control and the MF patient (Figure 5A), while more proliferating keratinocytes (the basal cells in the epidermis) were visualized at the junction between the epidermis and dermis by staining with Ki-67 in the MF sample, compared with the NS control sample (Figure 5B). Finally, vimentin, collagen I, and CD45 revealed the localization of the immune cells in the epidermis and dermis; in the latter, mostly enriched nearby blood vessels were identified by expression of CD31 and αSMA for both samples (Figure 5C,D). In addition, we found several subregions with more infiltrating immune cells in the MF sample compared with the NS control, based on CD45 staining (Figure 5D). To define the spatial organization of the immune cells in more detail for MF patients, we next focused on the analysis of lymphoid and myeloid cell populations in the MF tissue sections. Representative images are shown in Figure 5 in which we identified CD3^+^CD7^+^ T cells, abnormal CD3^+^CD7^−^ T cells and CD3^−^CD7^+^ ILCs (Figure 5E-01), CD3^+^CD4^+^ T cells and CD3^+^CD8^+^ T cells in the dermis, and CD3^+^CD4^−^CD8^−^ T cells in the epidermis (Figure 5E-02). In addition, analysis of the expression of CD45RA, FOXP3, and CD25 revealed that the large majority of T cells had a memory phenotype and allowed the distinction between CD45RA^−^ memory CD4^+^ T cells (Figure 5E-03) and FOXP3^+^CD25^+^ Tregs (Figure 5E-04). Moreover, various myeloid cells subsets were detected (Figure 5F) including CD11c^+^CD1a^+^CD1c^+^ DCs, CD11c^+^CD1a^−^CD1c^+^ DCs, and CD11c^+^CD1a^−^CD1c^−^ myeloid cells in the dermis; CD11c^−^CD1a^+^CD1c^+^ Langerhans-like cells (LCs) in the epidermis (Figure 5F-01); HLA-DR^+^CD163^+^ macrophages were present in the dermis (Figure 5F-02). In addition, we observed lower expression levels of HLA-DR in the epidermal LCs compared to dermal DCs (Figure 5F-03). Few HLA-DR^+^CD123^+^ pDCs-like cells were detected in the dermis (Figure 5F-04). Importantly, we observed prominent co-localization of CD4^+^ T cells with both CD1a^+^CD1c^+^ and CD1a^−^CD1c^+^ DCs in cellular aggregates just below the epidermis. Thus, by this approach, we were able to identify and visualize the presence and distribution of various lymphoid and myeloid immune cell subsets within a single tissue section simultaneously.

We next analyzed five additional early-stage MF patients, which revealed similar clusters of immune cells just below the epidermis in all MF patients (Figure 6). Appendix A provide an overview of the individual marker stains for all MF patients. The substantial heterogeneity of immune cells was observed among the MF patients. While CD4^+^ T cells were the most abundant lymphoid cells in patients 105MF, 109MF, and 113MF, CD8^+^ T cells were more abundant in patient 87MF and CD4^−^CD8^−^ T cells in patient 108MF (Figure 6A). Moreover, myeloid cells were virtually absent from the lymphoid cell aggregate in patient 87MF (Figure 6B), while co-localization of lymphoid cells and myeloid cells, in particular CD11c^+^CD1a^+^CD1c^+^ DCs, was observed in patients 105MF and 109MF and to a lesser extent in patients 108MF and 113MF (Figure 6C). Apart from patient 87MF, multiple types of antigen-presenting cells were identified in all other patients based on differential expression of CD11c, CD1a, and CD1c.

Together, this provides evidence that clusters of lymphoid and myeloid cells were found in the majority of early-stage MF patients. Moreover, it reveals substantial heterogeneity in both the lymphoid and myeloid compartments within and among patients.

## 4. Discussion

In the current study, we analyzed skin biopsies from early-stage MF patients and healthy controls to characterize the complexity of the immune compartment using high-dimensional single-cell suspension mass cytometry and imaging mass cytometry. Compared with conventional flow cytometry and immunohistochemistry, these mass cytometry-based techniques offer the opportunity to detect up to 40 cellular markers simultaneously, thus allowing a high-resolution analysis of the immune compartment in the tissue context.

Previous studies found that early stages of MF were characterized by the presence of a small number of neoplastic cells together with an extensive inflammatory infiltrate composed of multiple types of immune cells [21,22]. These observations have fueled the notion that this inflammatory response may contribute to the persistence and progression of MF lesions. In the present study, we identified phenotypically distinct subsets in both the CD4^+^ T-cell and myeloid cell compartment that were shared by most patients. In addition, we found distinct CD4^+^ T-cell subsets that revealed an individual pattern, potentially representing a unique response of the tumor cells to the tumor microenvironment. In addition, substantial numbers of CD4^+^ T cells co-localized with both CD1a^+^CD1c^+^HLA-DR^+^ and CD1a^−^CD1c^+^HLA-DR^+^ DCs in the dermis in situ. In this respect it is striking that the dominant presence of particular CD4^+^ T-cell clusters coincided with an elevated number of phenotypically distinct myeloid cells in some of the patients, suggesting that interactions between these CD4^+^ T cells and myeloid cells may play a prominent role in disease control. Here, it is of note that the composition, shape, and organization of the lymphoid–myeloid cell aggregates differed substantially among patients. Future studies need to be performed to determine if this relates to disease progression and/or may have implications for therapy.

Previously, studies focusing on Treg-like cells suggested that FOXP3^+^ Tregs have a tumor suppressive role in the pathogenesis of MF/SS, but the results have been discordant and conflicting [23,24,25]. In our single-cell mass cytometry analysis, clusters of Treg-like cells (CD25^+^CD45RO^+^CD27^+^) were found in all patients. This observation was confirmed using imaging mass cytometry where CD25^+^FOXP3^+^CD4^+^ T cells were found to be present in all investigated MF patients. Moreover, tumor-associated macrophages have been shown to generate an immunosuppressive tumor microenvironment by recruiting Tregs, myeloid cells, and the production of macrophage-related chemokines and angiogenic factors [26,27,28]. In patient 71MF, an increased number of CD163^+^ macrophages were observed (Figure 4). Therefore, in future studies, we will focus on the analysis of the interactions of Tregs with myeloid and tumor cells in the tissue context. Here, future studies investigating the therapeutic effect of IFN-alpha and -gamma for immunomodulation of tumor-associated macrophages might be of particular interest in such patients.

Finally, the PD-1/PD-L1 axis plays a central role in attenuating the immune response and antitumor immunity [29,30] and has also emerged as a central tumor suppressor in T-cell lymphomas [31]. We observed that PD-1 expression was higher in patients 71MF, 66MF, and 59MF (Figure 3). However, due to the short follow-up time, we could not observe a correlation between PD-1 expression and the disease course. Future studies should investigate if the presence of PD-1^+^CD4^+^ T cells correlates with progression to tumor stage disease. The therapeutic potential of PD-1 targeting therapy in these patients should be explored by investigating the factors driving PD-1 expression and functional consequences of PD-1 expression by CD4^+^ T cells as well.

In recent years, significant heterogeneity was observed between CTCL patients. Substantial clonotypic heterogeneity of skin- and blood-derived malignant T cells was observed by combining T-cell receptor clonotyping with cell surface marker profiling [32,33,34]. Moreover, genetic heterogeneity among and within CTCL patients was observed by TruSeq targeted RNA gene expression analysis [35]. These observations underscore the need to take into account the patients’ individual malignant profiles for effective therapy of CTCL.

Moreover, among the various single-cell techniques, cellular indexing of transcriptomes and epitopes by sequencing (CITE-seq) is a multimodal approach allowing simultaneous quantification of single-cell transcriptomes and surface proteins based on oligonucleotide-labeled antibodies of the same single cells [36]. Comparison of scRNA-seq, CyTOF, and CITE-seq analyses, however, reveals discrepancies with the highest abundance of the T-cell population in scRNA-seq analysis, followed by CyTOF and the lowest abundance in CITE-seq [37], pointing towards the need for further studies.

For future studies it would also be important to further optimize the antibody panels of single-cell CyTOF for the detection of malignant T cells combined with detection of co-stimulatory signals (e.g., CD80) [38], immune modulatory signals (e.g., CD137, CD134, and CTLA4) [39,40,41], T-cell exhaustion (e.g., ICOS) [42], and cytokine and chemokine receptors [43,44] to further dissect the tumor microenvironment. In addition, while in the present study, we used the MCD™ viewer software to visualize the imaging mass cytometry data, this could be complemented with cell segmentation approaches based on the identification of nuclei to aid in the visualization of IMC data [45,46]. Moreover, ImaCytE [47] and histoCAT [48], allow for downstream imaging mass cytometry analysis to identify and quantify cell–cell interactions [49,50]. Collectively, this allows for a further in-depth investigation of cellular interactions in skin tissues. Although the current study with a limited number of MF patient needs to be expanded to confirm and extend the observations, we nonetheless demonstrate the ability to detect immune cell profile patterns in single-cell suspensions of skin biopsies and to visualize the spatial network in the tissue context. The identification of prominent cellular aggregates between CD4 T cells and myeloid cells in the dermis with a patient-unique cellular composition provides a framework for improving mycosis fungoides diagnosis and development of treatment tailored to the characteristic features of these aggregates in individual patients.

## Figures and Tables

**Figure 1 cells-11-01062-f001:**
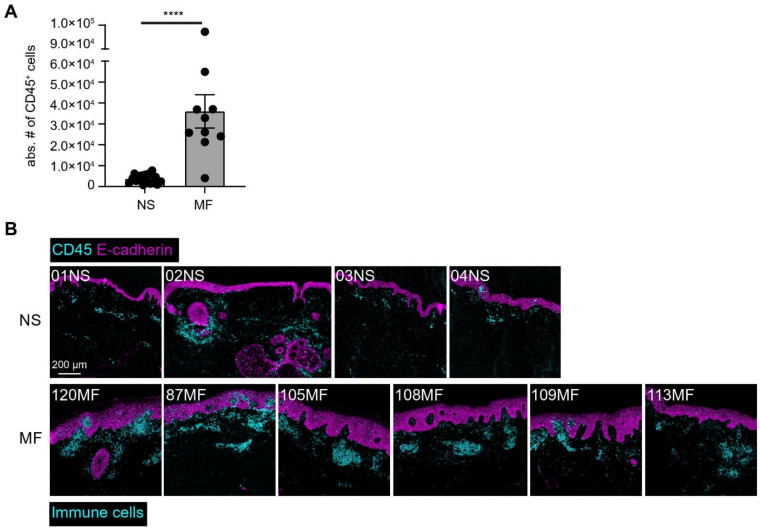
Immune cells quantification and spatial localization of NS and MF skin biopsies by mass cytometric analysis. (**A**) Live single CD45^+^ absolute cell numbers acquired for 17 normal skin biopsies (NS) and 10 MF skin biopsies by single-cell mass cytometry. **** *p* < 0.0001, using the Mann–Whitney U test. Error bars show the means ± SEM. (**B**) IMC images of the investigated NS control and MF samples showing the overlay of CD45 (cyan) and E-cadherin (magenta).

**Figure 2 cells-11-01062-f002:**
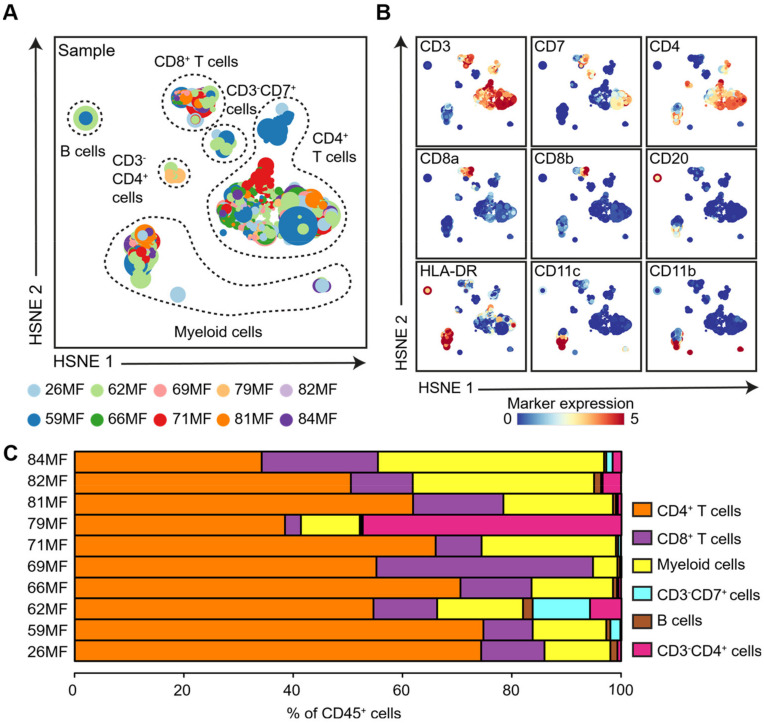
HSNE analysis revealed major immune lineages across MF patients. (**A**,**B**) HSNE embedding showing 6.9 × 10^4^ landmarks representing all immune cells (3.6 × 10^5^ cells) isolated from fresh skin biopsies of MF patients (*n* = 10) at the overview level. Each dot represents an HSNE landmark, and the size of the landmark indicates the number of cells that each landmark represents. Colors represent the individual MF patients (**A**) and the related expression level of indicated immune markers (**B**). (**C**) The composition of the major immune lineage populations of CD45^+^ cells in an individual MF patient is represented by horizontal bars, where the colored segment lengths represent the proportion of cells as a percentage of CD45^+^ cells in the sample. Colors represent the different major immune lineage populations.

**Figure 3 cells-11-01062-f003:**
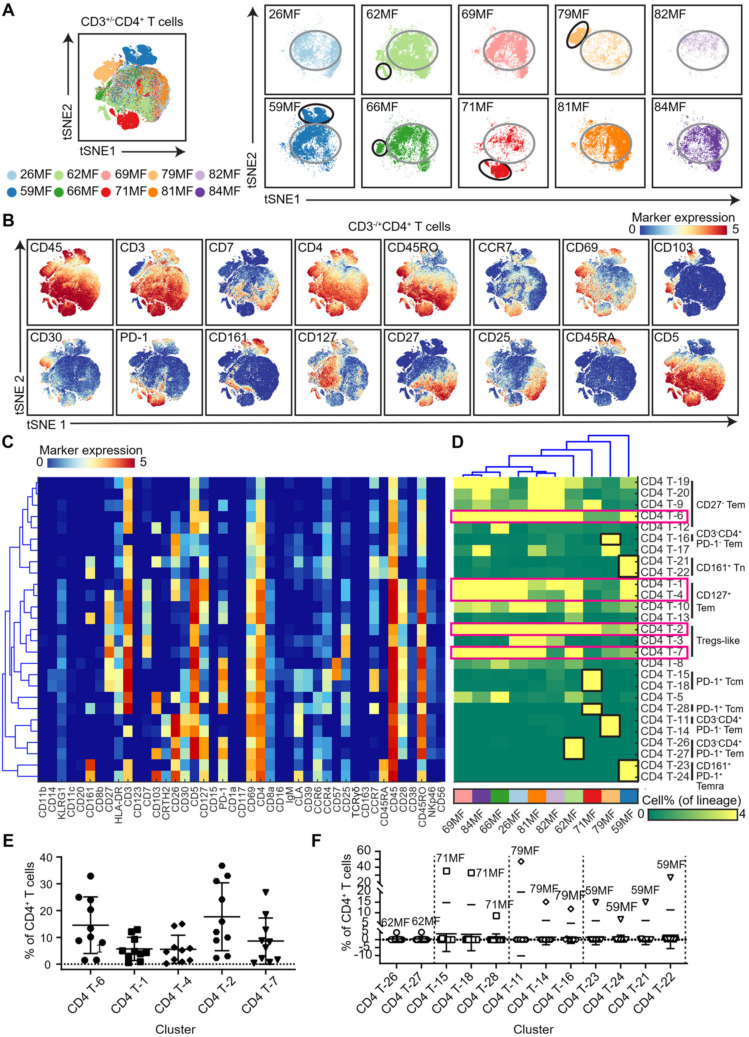
Identification of phenotypically distinct clusters in the CD3^+/−^CD4^+^ T-cell compartments across the MF samples. (**A**) A collective t-SNE was performed on CD4^+^ T cells (CD3^+^CD4^+^ T cells and CD3^−^CD4^+^ T cells) and stratified for samples (*n* = 10). In total, 2.3 × 10^5^ CD4^+^ T cells were analyzed in the plots. The large encircled cluster of cells (encircled in gray) harbored cells exhibiting a similar phenotype in all patients. The smaller cell clusters encircled in black were patient-unique and likely contained the abnormal cell populations in these patients. (**B**) Relative expression levels of the indicated immune markers. Colors represent different levels of marker expression. (**C**) Heatmap showing the median of the marker expression values for the clusters identified and hierarchical clustering thereof. (**D**) Heatmap showing the corresponding cell frequencies of identified clusters of total CD4^+^ T cells in each sample. Colors represent different MF samples as indicated below the heatmap. The dendrogram shows the hierarchical clustering of the samples. The clusters of CD4^+^ T cells highlighted by pink boxes were shared by the majority of the MF patients. The clusters of cells highlighted by black boxes were unique for the individual patients. (**E**) Quantification of the shared CD4^+^ T-cell cluster frequencies among the samples in (**C**). Cluster IDs corresponded to the ones in (**C**). (**F**) Quantification of the patient-unique CD4^+^ T-cell cluster frequencies among the samples in (**C**). Cluster IDs corresponded to the ones in (**C**).

**Figure 4 cells-11-01062-f004:**
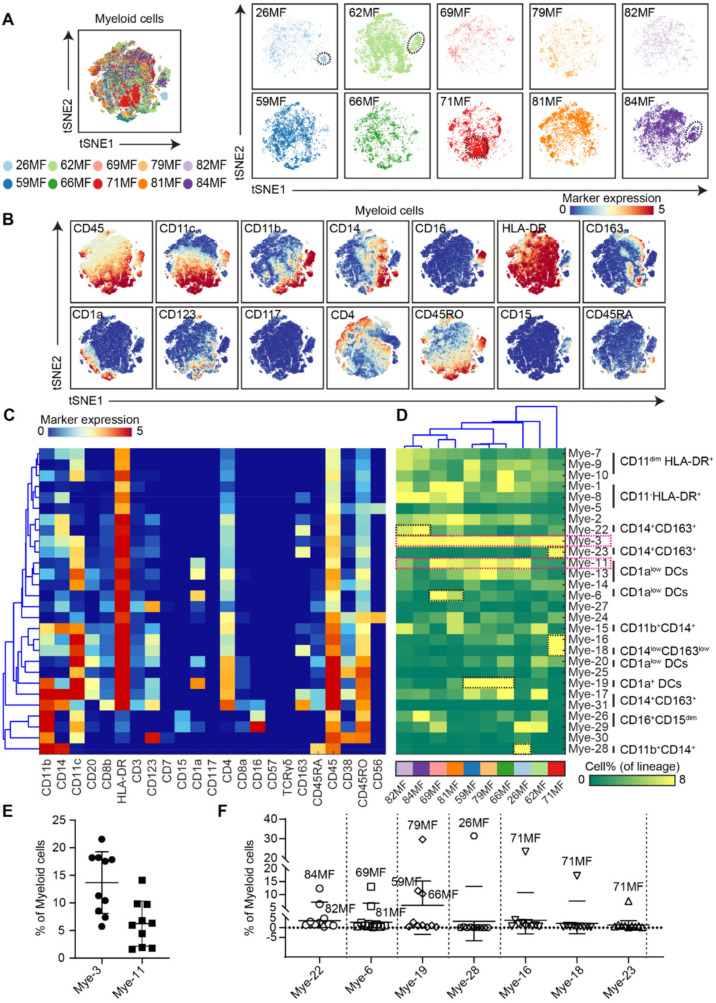
Identification of phenotypically distinct clusters in the myeloid cell compartment across MF samples. (**A**) A collective t-SNE was performed on myeloid cells and stratified for samples (*n* = 10). The plots show, in total, 6.1 × 10^4^ myeloid cells. The islands encircled in black are the unique-patient clusters of myeloid cells. (**B**) Relative expression level of the indicated markers. Colors represent the different levels of the marker expression. (**C**) Heatmap showing the median of the marker expression values for the clusters identified and hierarchical clustering thereof. (**D**) Heatmap showing the corresponding cell frequencies of the identified clusters as a percentage of total myeloid cells in each sample. Colors represent different samples as indicted below the heatmap. The dendrogram shows the hierarchical clustering of the samples. The clusters of the myeloid cells highlighted in the pink boxes were shared by the majority of MF patients. The clusters of myeloid cells highlighted in the black boxes are patient-unique. (**E**) Quantification of the shared myeloid cell cluster frequencies among the samples in (**C**). Cluster IDs corresponded to the ones in (**C**). (**F**) Quantification of the patient-unique myeloid cell cluster frequencies in (**C**). Cluster IDs corresponded to the ones in (**C**).

**Figure 5 cells-11-01062-f005:**
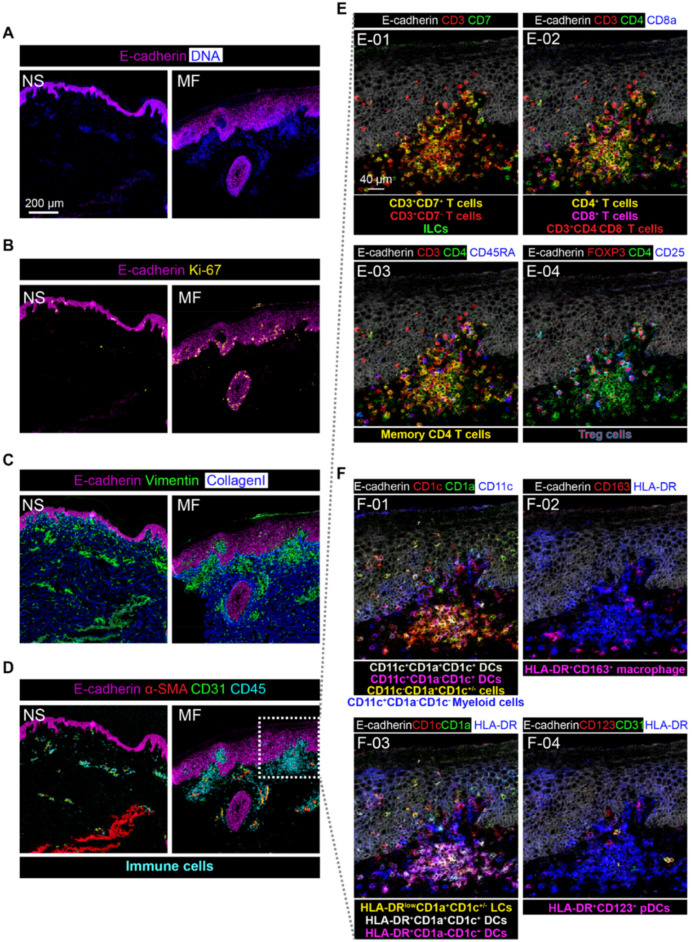
Visualization of the structure and the spatial distribution of the immune and stromal cell subsets in a single ROI in skin tissue by imaging mass cytometry: (**A**–**D**) representative mass cytometry images of an NS and an MF skin sample showing the overlay of (**A**) E-cadherin (colored in magenta) and DNA (colored in blue); (**B**) E-cadherin (colored in magenta) and Ki-67 (colored in yellow) to identify proliferating keratinocytes (E-cadherin^+^Ki-67^+^); (**C**) E-cadherin (colored in magenta), vimentin (colored in green), and collagen I (colored in blue) to distinguish the epidermis and dermis; (**D**) E-cadherin (colored in magenta), α-SMA (colored in red), CD31 (colored in green), and CD45 (colored in cyan) to show the location of CD45^+/dim^ immune cells; (**E**) identification of T-cell and ILCs subsets for the MF sample: (**E-01**) CD3^+^CD7^+^ T cells, CD3^+^CD7^−^ T cells, and ILCs (CD3^−^CD7^+^); (**E-02**) CD4^+^ T cells (CD3^+^CD4^+^), CD8^+^ T cells (CD3^+^CD8^+^), and CD4^−^CD8^−^ T cells (CD3^+^CD4^−^CD8^-^); (**E-03**) memory CD4^+^ T cells (CD3^+^CD4^+^CD45RA^−^); (**E-04**) regulatory T cells (Tregs, CD3^+^CD4^+^CD25^+^FOXP3^+^). (**F**) Identification of myeloid cell, dendritic cells, and macrophage subsets from the MF sample: (**F-01**) different dendritic cell subsets based on different expression levels of CD11c, CD1a, and CD1c; (**F-02**) CD163^+^ macrophage (HLA-DR^+^CD163^+^); (**F-03**) epidermal Langerhans cells (LCs, HLA-DR^dim^CD1a^+^ CD1c^+^) and dendritic cells (DCs); (**F-04**) plasmacytoid dendritic cell-like cells (pDC-like, HLA-DR^+^CD123^+^).

**Figure 6 cells-11-01062-f006:**
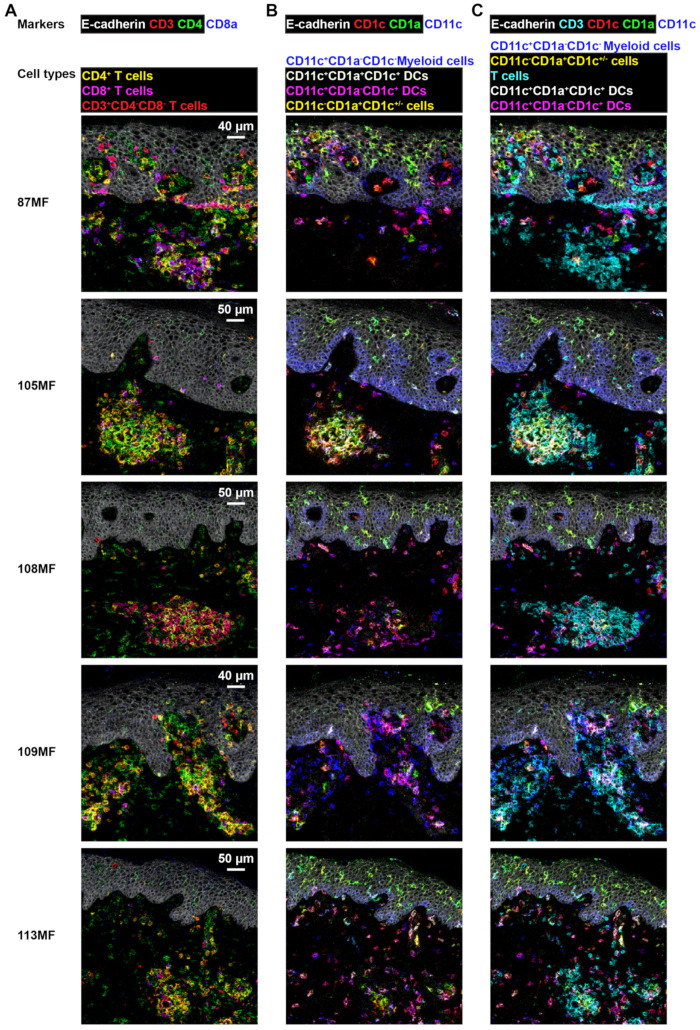
Detection of cell–cell interaction by combining T-cell markers with myeloid cell markers in skin biopsies of five additional MF patients: (**A**) visualization of CD4^+^ T cells, CD8^+^ T cells, and CD4^−^CD8^−^ T cells; (**B**) visualization of multiple types of antigen-presenting cells (APCs) by the overlay of CD1c (colored in red), CD1a (colored in green), and CD11c (colored in blue); (**C**) the overlay of CD3 (colored in cyan), CD1c (colored in red), CD1a (colored in green), and CD11c (colored in blue) shows the distribution of the myeloid and T-cell populations and the complex interactions among them.

## Data Availability

All data that support the findings of this study are available from the corresponding author upon reasonable request.

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
