# Peer review of "Mass Cytometric Analysis of Early-Stage Mycosis Fungoides"

_cells, 2022, doi:10.3390/cells11071062_

Round 1
Reviewer 1 Report
Early stages of mycosis fungoides (MF) present as inflamed skin patches or plaques covering a limited area of the skin and normally follows an indolent course. However, in a subset of patients the cutaneous lesions progress into tumors and the disease may spread to the lymph and internal organs with fatal consequences. Early MF skin lesions are characterized by the presence of a small population of malignant T cells admixed with a large population of benign immune cells. Evidence suggests that the composition of the benign immune cell compartment as well as interactions between malignant T cells and benign immune cells play an important role in the diseases pathogenesis. Nevertheless, there have been few studies comprehensively profiling the immune cell compartment and its tissue distribution in early MF lesions.
In the present manuscript, the authors for the first time apply high-dimensional single cell suspension and imaging mass cytometry to gain a deeper understanding of the composition, distribution and spatial interactions of the immune cell compartment in early MF lesions. Accordingly, they address a highly relevant research question in the field in a timely manner. The authors find patient-unique cell clusters in both the CD4+ and myeloid compartment but also distinct cell clusters that are shared by most patients. In addition, they observe clusters of CD4 T cells and multiple types of DCs in the dermis supporting that interactions between these cell types foster the disease pathogenesis. These findings contribute to our understanding of the composition and distribution of immune cells in early MF lesions and, in concert with recent studies, demonstrate substantial heterogeneity between CTCL patients.
The manuscript and data are presented in a clear and well-structured manner. The experimental design is rational and the methods adequately described except for a few ambiguities (see below). Furthermore, reasonable conclusions are drawn from the obtained results and the limitations of the study, such as the limited number of patients and short follow-up period, are appropriately discussed. Accordingly, the reviewer only have some minor comments to enhance the clarity and interpretation of the manuscript.
Minor comments:
Introduction
1. There is a writing mistake in line 56 where it states “… the malignant T-cells cell…”. Here “cell” should be deleted.
Material and methods
2. In line 85 it is written that “A total of 16 skin biopsies from 29 patients…”. The reviewer assumes that this is a typing mistake which should be corrected. If not, it should be clarified why only 16 biopsies from 29 patients were analyzed and what the selection/inclusion criteria were.
3. While it appears that all analyzed biopsies were taken before treatment, it is not completely clear. As prior treatment could have substantial impact on the tumor cells and composition of the local immune infiltrate, it is suggested that the authors explicitly state that the patients had not received prior therapy at the time the biopsies were taken. In case some patients should have received prior treatment at the time the biopsy was taken, this information should be included in Table S1.
4. It would be informative to include sex and age characteristics of the healthy donors in either the text or a small supplementary table.
Results & figures
5. Besides the 10 MF biopsies, it appears that 17 NS biopsies were analyzed by single-cell mass cytometry. However, the data obtained from the NS biopsies are only used to show the live single CD45+ absolute cell number acquired in Figure 1A and not described in further detail. If there were too few cells in the NS samples - or other technical issues preventing further analysis - this should be noted in the methods or results. Otherwise, if the data are available, it would be of interest to include a more detailed analysis of the NS samples (for example in the supplementary Figures) to give an impression of the which clusters of immune cells the authors find in normal skin and how much these vary between donors. This could add to the understanding and perspective of the MF single cell mass cytometry data.
6. In line 231-233 it is written that “Visual inspection of the heatmap indicated that the clustering of the 6 samples was to a large extent due to the sharing of clusters CD4 T-2, CD4 T-6 and to a lesser extent CD4 T-1 and CD4 T-4 (Figure 3D, pink boxes)”. Looking at Figure 3D, CD4 T-7 is also highlighted by a pink box and appears to be shared by 5 of the 6 samples. Therefore, it is suggested to either include CD4 T-7 together with CD4 T1- and CD4 T-4 in line 233 or remove the pink box in Figure 3D. If CD4 T-7 is included, it could be argued that CD4 T-19 should also be included.
7. In Figure 4D, it states CD14+CD163- besides Mye-17 and Mye-31. However, looking at Figure 4C it appears that Mye-17 and Mye-31 express CD163.
8. It seems that 4 NS samples have been analyzed by Imaging Mass Cytometry. Yet, only a few representative images from a single NS patient are shown. If possible, it would be informative to provide an overview of the individual markers strains for all the analyzed NS samples in the supplemental Figures, as it has been done for the MF patients.
Discussion
9. As heterogeneity is a central theme of the study, the manuscript would gain from an expansion of the discussion with a small paragraph on disease heterogeneity between patients and even within malignant sub-clones in each individual patient and include recent references on the subject in general - like Litvinov IV et al 2017 and Iyer et al 2019 (c.f. below) and the studies on heterogeneity utilizing single cell RNA seq studies like Buus et al 2018; Hamrouni et al 2019; Iyer et al 2020; and Herrera et al 2020 (c.f. below). Moreover, a brief discussion or reflections on strength/weaknesses/supplementation of CyTOF versus CITEseq could be of interest to readers.
Gene expression analysis in Cutaneous T-Cell Lymphomas (CTCL) highlights disease heterogeneity and potential diagnostic and prognostic indicators. Litvinov IV, et al .Oncoimmunology. 2017 Mar 17;6(5):e1306618.
Clonotypic heterogeneity in cutaneous T-cell lymphoma (mycosis fungoides) revealed by comprehensive whole-exome sequencing. Iyer A, et al .Blood Adv. 2019 Apr 9;3(7):1175-1184.
Single-cell heterogeneity in Sezary syndrome. Buus TB, et al Blood Adv. 2018 Aug 28;2(16):2115-2126.
Clonotypic Diversity of the T-cell Receptor Corroborates the Immature Precursor Origin of Cutaneous T-cell Lymphoma. Hamrouni A, Fogh H, Zak Z, Ødum N, Gniadecki R. Clin Cancer Res. 2019 May 15;25(10):3104-3114.
Branched evolution and genomic intratumor heterogeneity in the pathogenesis of cutaneous T-cell lymphoma. Iyer A, Hennessey D, O'Keefe S, Patterson J, Wang W, Wong GK, Gniadecki R. Blood Adv. 2020 Jun 9;4(11):2489-2500
Multimodal single-cell analysis of cutaneous T-cell lymphoma reveals distinct subclonal tissue-dependent signatures. Herrera A et al. Blood. 2021 Oct 21;138(16):1456-1464.
Author Response
We would like to thank the reviewer for these positive comments.
Please see the attachment including our responses of point by point.

Reviewer 2 Report
The authors used suspension and imaging mass cytometry to interrogate multiple normal and Mycosis Fungoides skin lesions. While the sample sizes were small and the numbers appear to be confused by the authors some interesting data resulted. 16 skin from confirmed MF 29 patients and 17 normal. However, the break down for suspension and imaging is where the numbers get confusing. For suspension 10 MF and 17 normal and for imaging 6 MF and 4 normal. The issue is that the suspension used fresh biopsies and the imaging the snap-frozen, meaning that 21 normal samples were used, not the 17 reported in the first sentence (line 86).
The authors went to some length to analyze the suspension data clustering the data into the immune populations and then further subclustering the data by cell type and found some very interesting details where t-cells lost their CD3 and CD7 expression. They went deep into the cells identified and the variances in the numbers found and even pointed out the extreme variance in the CD45 population between normal and the MF samples. However, when the authors approached their imaging mass cytometry data the paper becomes completely descriptive as they didn't apply any analytical process beyond observation, which can be challenging given that the data is 16 bit and they are attempting to characterize 36 markers by eye, likely on an 8 bit monitor. This is a substantial flaw in the data given the novel populations described in the suspension data. The authors are clearly aware of the analytical techniques as they discuss several options in the discussion section alluding to what could be done on the next manuscript. I believe this would have benefited from several samples being used by both techniques, not merely one set to suspension an a second set to imaging. The imaging seems to be an afterthought to the manuscript. The collected images should have been tissue and cell segmented and the descriptive terms replaced with cell numbers and distances between tissue areas and between phenotypes of interest. Given the interesting observations from the suspension data I am asking where are these cd3- and cd7- t-cells located in the tissues, especially in patient 79MF. I believe these details need to be addressed for the manuscript to move forward. The imaging is lovely, but merely qualitative in it's inclusion.
Author Response

(The authors gave the same response as above.)

Round 2
Reviewer 2 Report
At issue is the use of MCD Viewer for "analysis" of the imaging data. MCD viewer sets an arbitrary value for the dynamic range (display of the data from the darkest black value to white or brightest value). These settings are not save-able outside of a researcher recording them. This means when you load image 1 and image 2 these settings will be different and affect the way the eye perceives a positive and a negative value. As the researcher is reporting on phenotypes that disappear in part of the samples (CD3 & CD7) without standard presentation of the images this is hard to appreciate and evaluate the images. The researchers do not report the dynamic ranges collected for each, nor do they provide a table of the text files that reports them when images are exported from MCD viewer. This makes even their visible comparison a challenge and must be improved prior to publication. If a qualitative assessment is all the authors desire in this publication they must at minimum standardize the dynamic range values of each channel to the min and max values for that channel across all the images so that brightness and dimness and thus protein expression are comparable.
Also, the authors state irregular cell shapes as the issue to segmenting and more fully analyzing the data. Please know brain tissue is regularly published containing Astrocytes, which features a very irregular shape. Astrocytes are managed, so this justification is not appropriate.
Author Response
We would like to thank the reviewer for these comments.
Please see the attachment including our responses of point by point.
